# Murine HSCs contribute actively to native hematopoiesis but with reduced differentiation capacity upon aging

**Petter Säwen[1], Mohamed Eldeeb[1], Eva Erlandsson[1], Trine A Kristiansen[1], Cecilia Laterza[2,3], Zaal Kokaia[2,3], Göran Karlsson[1,2,3], Joan Yuan[1,2,3], Shamit Soneji[1,2,3], Pankaj K Mandal[4,5], Derrick J Rossi[4,5], David Bryder[1,2,3,6]\***

[1]Division of Molecular Hematology, Department of Laboratory Medicine, Medical Faculty, Lund University, Lund, Sweden; [2]StemTherapy, Lund University, Lund, Sweden; [3]Lund Stem Cell Center, Lund University, Lund, Sweden; [4]Department of Stem Cell and Regenerative Biology, Harvard University, Cambridge, United States; [5]Program in Cellular and Molecular Medicine, Division of Hematology/Oncology, Boston Children's Hospital, Massachusetts, United States; [6]Sahlgrenska Cancer Center, Gothenburg University, Gothenburg, Sweden

**Abstract** A hallmark of adult hematopoiesis is the continuous replacement of blood cells with limited lifespans. While active hematopoietic stem cell (HSC) contribution to multilineage hematopoiesis is the foundation of clinical HSC transplantation, recent reports have questioned the physiological contribution of HSCs to normal/steady-state adult hematopoiesis. Here, we use inducible lineage tracing from genetically marked adult HSCs and reveal robust HSC-derived multilineage hematopoiesis. This commences via defined progenitor cells, but varies substantially in between different hematopoietic lineages. By contrast, adult HSC contribution to hematopoietic cells with proposed fetal origins is neglible. Finally, we establish that the HSC contribution to multilineage hematopoiesis declines with increasing age. Therefore, while HSCs are active contributors to native adult hematopoiesis, it appears that the numerical increase of HSCs is a physiologically relevant compensatory mechanism to account for their reduced differentiation capacity with age.
DOI: https://doi.org/10.7554/eLife.41258.001

**\*For correspondence:**
david.bryder@med.lu.se

**Competing interests:** The authors declare that no competing interests exist.

## Introduction

HSC-derived hematopoiesis has usually been studied in the setting of transplantation (*Benz et al., 2012*; *Biasco et al., 2016*; *Dykstra et al., 2007*; *Lu et al., 2011*; *Rundberg Nilsson et al., 2015*; *Wu et al., 2014*), an experimental paradigm that has been the foundation of hematopoietic research for decades (*Siminovitch et al., 1963*; *Till and McCulloch, 1961*) and which has established hallmark properties of HSCs such as multi-potency and self-renewal. However, while the transplantation assay has provided key insights, not the least with relevance for the clinical use in bone marrow (BM) transplantation, it might not accurately reflect the contribution of HSCs to ongoing and unperturbed steady state hematopoiesis. This is because transplantation is conducted under highly non-physiological conditions wherein HSCs are forced to proliferate to rebuild an entire hematopoietic hierarchy in a myeloablated bone marrow micro-environment. Therefore, there is a need to approach HSC biology also in more unperturbed settings.

While the overall structure of hematopoiesis is rather well established (*Bryder et al., 2006*), the degree by which HSCs contribute to adult hematopoiesis in the steady state is more unclear. This includes whether the proposed differentiation routes for the hematopoietic lineages are obligatory,

**eLife digest** As far as we know, all adult blood cells derive from blood stem cells that are located in the bone marrow. These stem cells can produce red blood cells, white blood cells and platelets – the cells fragments that form blood clots to stop bleeding. They can also regenerate, producing more stem cells to support future blood cell production. But, our understanding of the system may be incomplete.

The easiest way to study blood cell production is to watch what happens after a bone marrow transplant. Before a transplant, powerful chemotherapy kills the existing stem cells. This forces the transplanted stem cells to restore the whole system from scratch, allowing scientists to study blood cell production in fine detail. But completely replacing the bone marrow puts major stress on the body, and this may alter the way that the stem cells behave. To understand how adult stem cells keep the blood ticking over on a day-to-day basis, experiments also need to look at healthy animals.

Säweń et al. now describe a method to follow bone marrow stem cells as they produce blood cells in adult mice. The technique, known as lineage tracing, leaves an indelible mark, a red glow, on the stem cells. The cells pass this mark on every time they divide, leaving a lasting trace in every blood cell that they produce. Tracking the red-glowing cells over time reveals which types of blood cells the stem cells make as well as provides estimates on the timing and extent of these processes.

It has previously been suggested that a few types of specialist blood cells, like brain-specific immune cells, originate from cells other than adult blood stem cells. As expected, the adult stem cells did not produce such cells. But, just as seen in transplant experiments, the stem cells were able to produce all the other major blood cell types. They made platelets at the fastest rate, followed by certain types of white blood cells and red blood cells. As the mice got older, the stem cells started to slow down, producing fewer blood cells each. To compensate, the number of stem cells increased, helping to keep blood cell numbers up.

This alternative approach to studying blood stem cells shows how the system behaves in a more natural environment. Away from the stresses of transplant, the technique revealed that blood stem cells are not immune to aging. In the future, understanding more about the system in its natural state could lead to ways to boost blood stem cells as we get older.

DOI: https://doi.org/10.7554/eLife.41258.002

or whether alternative/complementary pathways exist. Furthermore, cells of the different hematopoietic lineages have not only distinct homeostatic functions and maintenance mechanisms (*Bando and Colonna, 2016*; *Dzierzak and Philipsen, 2013*; *Rodvien and Mielke, 1976*) but also display dramatically different lifespans (*Galli et al., 2011*; *Harker et al., 2000*; *Van Putten, 1958*; *Westera et al., 2013*). As a consequence, the rates by which separate adult-derived blood cell lineages must be replenished differ substantially. At the extreme end, certain hematopoietic cell types generated during the fetal period appear devoid of replenishment from adult progenitors, and rather rely on homeostatic proliferation for their maintenance (*Ginhoux and Guilliams, 2016*; *Kantor et al., 1995*).

Recent developments of transgenic mouse models that allow for identification (*Acar et al., 2015*; *Chen et al., 2016*; *Gazit et al., 2014*) and evaluation of HSCs biology have facilitated studies of native in vivo hematopoiesis (*Busch et al., 2015*; *Sawai et al., 2016*; *Sun et al., 2014*; *Wilson et al., 2008*). Using one such model, we recently revealed that most adult HSCs are highly quiescent, which is strikingly different in the transplantation scenario (*Säweń et al., 2016*). Other models have been used for lineage tracing from HSCs (*Busch et al., 2015*; *Sawai et al., 2016*; *Sun et al., 2014*). In one of these, lineage tracing was conducted via random genetic integration of an inducible transposable genetic element, leading to the proposition that native hematopoiesis involves a large number of actively contributing progenitor cell clones, which are only rarely shared among hematopoietic lineages (*Sun et al., 2014*). More common approaches for lineage tracing involve the use of cell type specific recombinases, that function to irreversibly mark a cell of interest and with time its descendants. While elegant and extensively used among developmental biologists, such approaches have only sparsely been applied to adult HSCs, and with seemingly contradictory results. Using a Tie2-driven model, Busch et al. concluded a substantial hematopoietic contribution/maintenance from progenitors rather than HSCs (*Busch et al., 2015*), which at least to some extent would appear

compatible with the results from *Sun et al. (2014)*. By contrast, Sawai et al. utilized a Pdzk1ip1-based CreERT2 system and suggested robust HSC labeling and hematopoiesis from adult HSCs (*Sawai et al., 2016*).

To try to assess these potential ambiguities, we here investigated the degree to which HSCs contribute to steady state adult hematopoiesis by using an inducible Fgd5-based HSC lineage tracing model (*Gazit et al., 2014*). We observed dramatic differences with regards to HSC contribution to adaptive immunity (slow) and the myeloerythroid lineages (fast), with HSCs contributing to the platelet lineage with the most rapid kinetics. The regeneration of terminal cell fates was closely mirrored at the level of each intermediate myeloerythroid precursor. These findings are consistent with adult HSCs as highly active contributors to multilineage hematopoiesis not only following transplantation, but also during the steady state. However, when approached in the situation of chronological aging, we noted diminished mature blood cell output from aged HSCs that could be traced to the first differentiation events from HSCs. These results suggest that the previously proposed fetal to adult switch (*Bowie et al., 2007*), in which HSCs alter their properties from more excessive proliferation/differatiation to a more dormant state in the adult, extends gradually throughout adulthood. As a consequence, the well-known numerical increase of HSCs with age (*Morrison et al., 1996*; *Rossi et al., 2005*; *Sudo et al., 2000*) appears to represent a physiologically relevant mechanism to account for reduced HSC differentiation with age.

## Results

### Fgd5-CreERT2-based lineage tracing allows for assessment of HSC contributions to unperturbed hematopoiesis

Using a transcriptome based screen of more than 40 different hematopoietic cell types, Fgd5 (FYVE, RhoGEF and PH domain containing 5) was identified as a HSC-expressed gene that is rapidly downregulated upon differentiation. That Fgd5 expression marks all HSCs was confirmed through functional studies using an Fgd5 knock-in reporter strain (*Gazit et al., 2014*). To further detail the HSC specificity of Fgd5, we first acquired transcriptome data from 11,581 individual lineage-marker negative, c-kit positive and CD45 positive bone marrow cells ($Lin^-kit^+$). The $Lin^-kit^+$ population contains a range of different immature hematopoietic progenitor cells (*Pronk et al., 2007*). Therefore, $Lin^-kit^+$ cells provided a benchmark to which other more defined/specific hematopoietic progenitor subsets could be compared. Next, we took advantage of an Fgd5 reporter strain in which a ZsGreen-2A-CreERT2 allele was knocked into the endogenous *Fgd5* locus (hereafter *Fgd5*$^{CreERT2/+}$ mice) (*Figure 1B*) (*Gazit et al., 2014*). We sorted either $Lin^-kit^+Fgd5^+$ cells (*Figure 1A* middle; 793 cells, $Fgd5^+$), or $Fgd5^+$ cells with a stringent $Lin^-kit^+Sca-1^+CD48^-CD150^+$ HSC phenotype (*Figure 1A* right, 519 cells, HSC-$Fgd5^+$). All $Fgd5^+$ and HSC-$Fgd5^+$ data were aggregated with the $Lin^-kit^+$ transcriptome data, which was followed by identification of the most significant gene vectors using principal component analysis (PCA). Data was then visualized using t-distributed stochastic neighbor embedding (tSNE) dimensionality reduction (*Figure 1A*). $Lin^-kit^+$ cells were extensively scattered across the two dimensions (*Figure 1A*, left), in agreement with the heterogeneity of these cells. By contrast, $Fgd5^+$ cells, regardless if sorted based on additional HSC markers, formed a distinct and highly overlapping cluster (*Figure 1A*, middle and right). This cluster localized to a region with very few cells when evaluating $Lin^-kit^+$ cells (*Figure 1A*, left, dotted area), emphasizing the HSC-specificity of the Fgd5 reporter and the low HSC frequency within the larger $Lin^-kit^+$ fraction.

We next generated a lineage tracing model by crossing *Fgd5*$^{CreERT2/+}$ mice to *Rosa26-Lox-Stop-Lox-Tomato* mice (hereafter *Rosa26*$^{lsl-Tomato/+}$) (*Figure 1B*). In this model, HSCs can be identified based on ZsGreen expression, while Tamoxifen administration leads to irreversible and heritable Tomato labeling of HSCs and, over time, their offspring (*Figure 1C*). To confirm the model, we evaluated Tomato label in HSC and BM progenitor cells 48 hr after a single injection (1x) of Tamoxifen. This revealed labeling of a fraction of candidate HSCs, with virtually no labeling in other c-kit$^+$ progenitor fractions (*Figure 1D* and *Figure 1—figure supplement 1*). This established HSC specific labeling and a relatively low differentiation rate of HSCs in steady state (*Säwén et al., 2016*; *Wilson et al., 2008*). To illustrate our ability to detect Tomato label in peripheral blood (PB) cells, we assessed Tomato expression in defined cell types from mice that had received Tamoxifen 8–48 weeks previously (*Figure 1D*, lower right). Complementary to immunophenotypic identification of

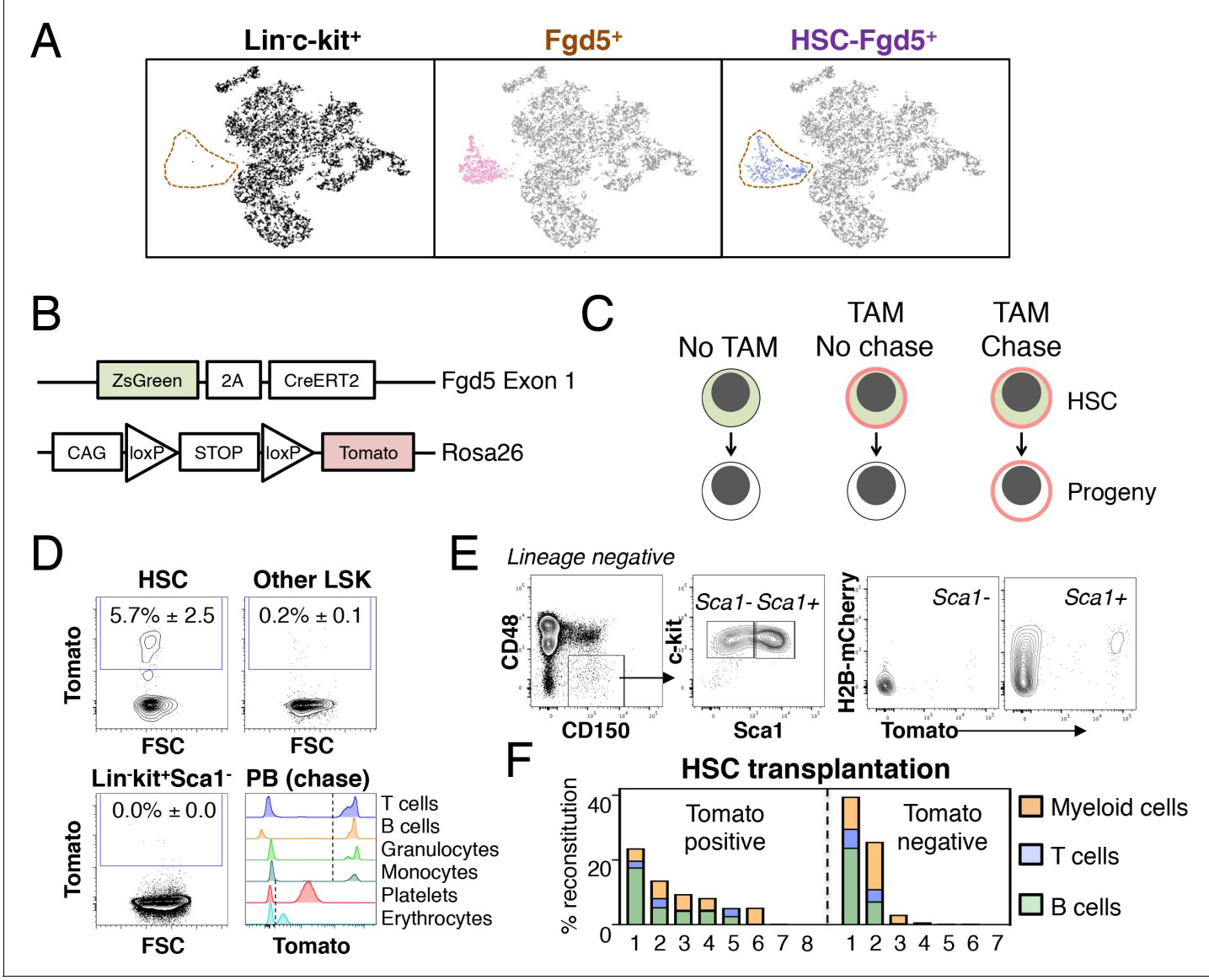

**Figure 1.** Fgd5-CreERT2 specifically labels HSCs and Fgd5-mediated label progresses throughout the hematopoietic system. (A) Lineage negative c-kit$^+$ cells (Lin$^-$c-kit$^+$, left), lineage negative c-kit$^+$ Fgd5$^+$ cells (Fgd5$^+$, middle) and lineage negative Fgd5$^+$c-kit$^+$Sca-1$^+$CD150$^+$CD48$^-$ cells (HSC-Fgd5$^+$, right) were isolated and subjected to single cell RNA-sequencing. The data was aggregated and visualized in a two-dimensional scatter plot after PCA and tSNE dimensionality reduction. Fgd5$^+$ cells are highlighted in pink (middle), Lin$^-$c-kit$^+$ cells are highlighted in black (left plot) and HSC-Fgd5$^+$ cells are highlighted in blue (right plot). The area that Fgd5$^+$ cells occupy in relation to the transcriptomes of Lin$^-$c-kit$^+$ cells and HSC-Fgd5$^+$ cells is marked by a dotted line (left and right plots). (B) Schematic representation of the $Fgd5^{CreERT2/+}$; $Rosa26^{lsl-Tomato/+}$ model. ZsGreen and CreERT2 are expressed from the Fgd5 locus and expression of a Tomato allele is driven by a CAG promoter from the Rosa26 locus and is preceded by a LoxP flanked STOP cassette. (C) Model description; HSCs selectively and continuously express ZsGreen in an Fgd5-dependent manner. Upon Tamoxifen (TAM) administration, HSCs express Tomato and expression of Tomato label is inherited by all progeny of Tomato-expressing HSCs. (D) Representative FACS plots showing Tomato label in BM HSPCs from $Fgd5^{CreERT2/+}$; $Rosa26^{lsl-Tomato/+}$ mice that were injected with Tamoxifen 48 hr prior to analysis. (D, lower right) Representative histograms depicting Tomato label in PB cells at various time points after the start of Tamoxifen administration from mice in *Figure 3B* (T cells 48 weeks, B cells 25 weeks, granulocytes and monocytes 8 weeks, platelets and erythrocytes 13 weeks). Numbers in FACS plots depict the mean % of Tomato labeled cells ± SD (n = 5) and dashed lines in histograms indicates the boundary for Tomato positivity. (E) FACS plots showing H2B-mCherry label retention and Tomato labeling in Lineage$^-$c-kit$^+$CD150$^+$CD48$^-$ and Sca1$^+$ or Sca1$^-$ cells from a representative mouse that had diluted H2B-mCherry label for 5 weeks and were injected with Tamoxifen 5 days prior to analysis (n = 3; 14–19 weeks old at analysis). (F) The fraction of donor-derived cells among different blood cell lineages was assessed in individual mice 16 weeks post-transplantation in recipients of 5 Tomato$^+$ (n = 8) or 5 Tomato$^-$ (n = 7) HSCs. Abbreviations: 2A, 2A self-cleaving peptide; CAG, CAG promoter; loxP, LoxP site.

DOI: https://doi.org/10.7554/eLife.41258.003

The following figure supplement is available for figure 1:

*Figure 1 continued on next page*

*Figure 1 continued*

**Figure supplement 1.** FACS gating strategies for identification of hematopoietic subsets.

DOI: https://doi.org/10.7554/eLife.41258.004

initially labeled BM cells as HSCs (*Figure 1D* and data not shown), we evaluated the proliferation history of Tomato labeled HSPCs 5 days after a pulse of Tomato labeling by evaluation of transgenic H2B-mCherry label retention (*Figure 1E*) (*Säwén et al., 2016*). Among HSCs, this revealed a strong correlation between a restricted proliferative history and Tomato labeling. Of note, a single dose of Tamoxifen was insufficient to label all candidate Fgd5-expressing HSCs (*Figure 1E* and data not shown).

Finally, to corroborate that Tomato labeled phenotypic HSCs are bona fide HSCs, we injected mice with Tamoxifen and isolated candidate Tomato positive and negative HSCs 48 hr later. Sorted cells were transplanted at limiting dilution (5 cells/mouse). This revealed long-term multilineage reconstitution in 5/8 recipients transplanted with Tomato$^+$ HSCs (*Figure 1F*).

## Fgd5-lineage tracing reveals that HSCs generate different types of hematopoietic progeny with distinct kinetics

Encouraged by the highly specific HSC label observed after Tamoxifen administration to *Fgd5$^{CreERT2/+}$*; *Rosa26$^{lsl-Tomato/+}$* mice (*Figure 1*), we next set out to perform label tracing studies of hematopoietic generation from HSCs. For this, we labeled cohorts of *Fgd5$^{CreERT2/+}$*; *Rosa26$^{lsl-Tomato/+}$* mice with one injection of Tamoxifen and chased groups of mice for different periods of time up to 83 weeks after labeling. At end point analyses, the fraction of Tomato$^+$ cells was determined in various hematopoietic compartments to assess the HSC contribution to progenitor pools and mature blood cell subsets (*Figure 1—figure supplement 1*). The frequencies of Tomato$^+$ cells for each investigated subset were next related to the fraction of Tomato labeled HSCs (*Figure 2A*, mean 13% ± 9%) in individual mice (*Figure 2B*, and *Figure 2—figure supplement 1*). The fraction of labeled HSCs was generally higher in mice analyzed beyond 4 days of chase compared to mice analyzed after shorter chase periods. However, no further increase in HSC labeling was noted after longer periods of chase (*Figure 2A*).

First, we investigated Tomato label progression into the immature lineage negative, Sca-1 positive and c-kit positive (LSK) compartment, fractionated further using the Slam markers CD48 and CD150 (*Kiel et al., 2005*) (*Figure 2B* and *Figure 1—figure supplement 1*). We used this approach to identify HSCs (LSKCD150$^+$CD48$^-$) and different multipotent progenitor fractions (MPPs: LSKCD150$^-$CD48$^-$, MPP$^2$: LSKCD150$^+$CD48$^+$, MPP$^{3/4}$: LSKCD150$^-$CD48$^+$).

LSKCD150$^-$CD48$^-$ MPPs are immature multipotent progenitors distinguished from HSCs by their limited self-renewal potential (*Kiel et al., 2005*; *Kiel et al., 2008*; *Ugale et al., 2014*). Of the evaluated progenitor subsets in our work, this subset was generated from HSCs with the fastest kinetics, with near equilibrium to HSC label reached already by 4 weeks (*Figure 2B*).

MPP$^2$ cells represent a rare subset of cells with more undefined lineage/developmental affiliations. This prompted us to first elucidate their developmental potential. First, we aimed to place these cells within a transcriptional framework established by other, more established, hematopoietic progenitors. For this, we obtained gene expression data from a panel of defined stem and progenitor cells and MPP$^2$ cells using a multiplexed qRT-PCR approach for 48 genes, selected to include cell surface markers, cell cycle regulators and transcription factors associated with hematopoiesis (*Supplementary file 2*). Visualization of this data using PCA revealed that MPP$^2$ cells clustered closely to Meg/E progenitors (*Figure 2C*). Consistent with a close association to the Meg/E lineages, short-term (6 days) culture experiments revealed a more robust generation of both megakaryocyte and erythroid containing colonies from MPP$^2$s compared to other LSK subsets (*Figure 2—figure supplement 2*). When investigating Tomato label progression, MPP$^2$ cells reached label equilibrium with HSCs after 32 weeks in 1x injected mice (*Figure 2B*).

MPP$^{3/4}$ cells lack, for the most part, Meg/E lineage potential (*Adolfsson et al., 2005*; *Arinobu et al., 2007*; *Pietras et al., 2015*; *Pronk et al., 2007*). MPP$^{3/4}$ cells acquired Tomato label with much slower kinetics compared to other LSK fractions (*Figure 2B*).

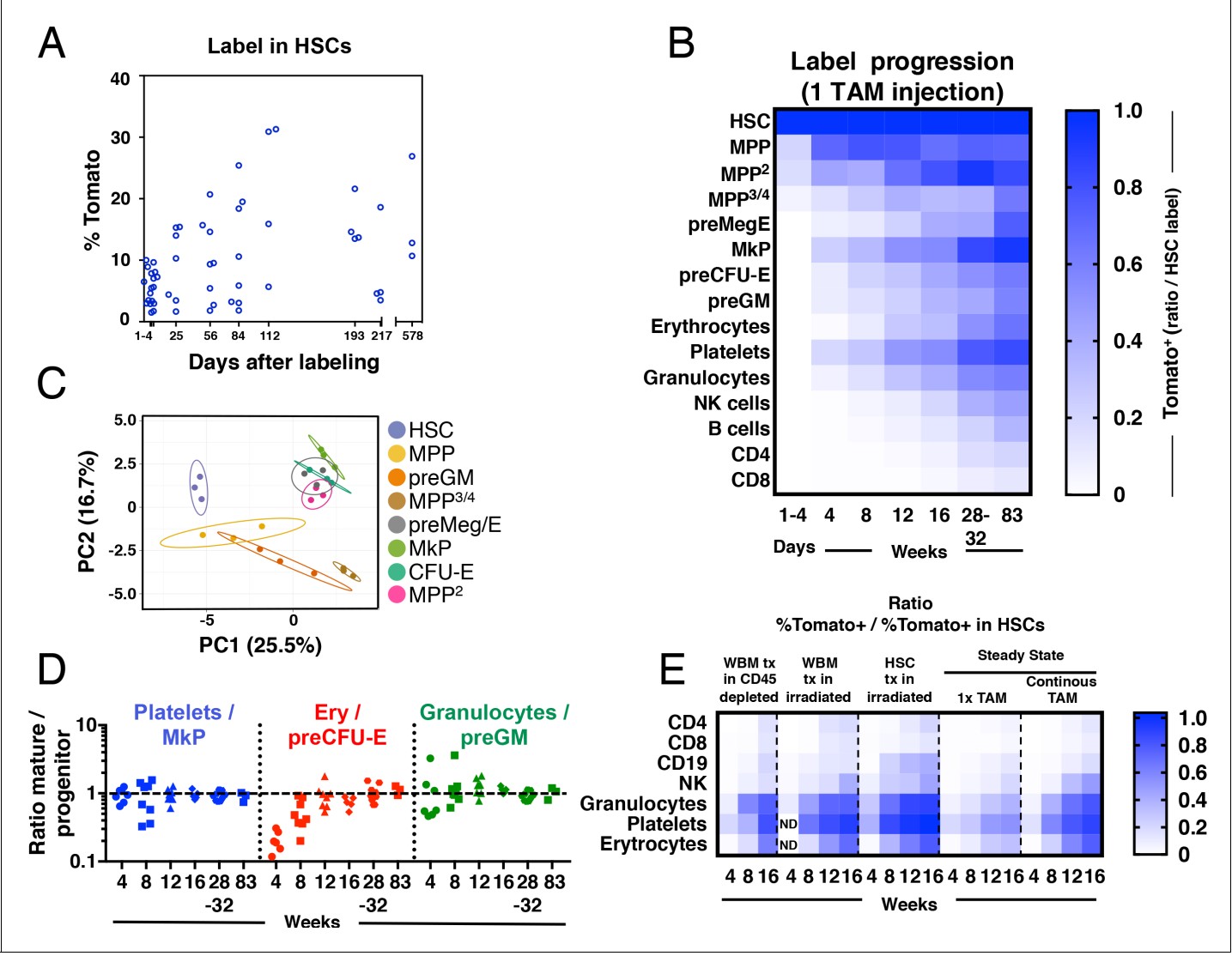

**Figure 2.** Fgd5-CreERT2 mediated lineage tracing reveals robust HSC contribution with distinct kinetics into hematopoietic cell subsets in steady state and after transplantation. Cohorts of *Fgd5^CreERT2/+; Rosa26^lsl-Tomato/+* mice were chased for up to 83 weeks after receiving one Tamoxifen injection before endpoint analysis of the % of Tomato⁺ cells in HSCs (A) and hematopoietic progenitor and PB cell subsets (B). The fraction of Tomato⁺ cells in each indicated hematopoietic cell subset was divided by the % Tomato label in HSCs in corresponding mice to determine the ratio of Tomato⁺ cells relative to HSCs in individual mice. The average ratio is plotted according to time after the Tamoxifen (TAM) injection and displayed in a heat map format. Mice were analyzed at 1–4 days (n = 18) and at 4 (n = 7), 8 (n = 8), 12 (n = 8), 16 (n = 4), 28 (n = 4), 32 (n = 5) and 83 (n = 3) weeks after the Tamoxifen injection. All mice were between 5 and 11 weeks old at the time of Tamoxifen injection. (C) PCA plot of multiplexed qRT-PCR data for 48 genes from triplicates of 10 cells from each of the indicated populations. Ellipses show an area where a new observation from the same group would position itself with a probability of 0.95. Numbers indicate the % of variance in the total data set that the respective PC explains. (D) For individual animals, the % of Tomato⁺ cells in PB cell types was divided by the % of Tomato⁺ cells in the indicated progenitor cell types and this value was plotted according to the duration of the chase period. Symbols represent individual mice (number of mice analyzed as in A and B). A dashed line is drawn to indicate an equilibrium-ratio of 1. (E) Ratios calculated as in B and plotted according to the number of weeks after transplantation or after the start of Tamoxifen administration for steady state/native mice (n = 3 for WBM tx in CD45 depleted, n = 5 for WBM tx in irradiated, n = 11 for HSC tx in irradiated, n = 4–8 for 1x TAM, n = 4–9 for continous TAM). All *Fgd5^CreERT2/+; Rosa26^lsl-Tomato/+* mice/cells were between 5 and 11 weeks old at the time of Tamoxifen administration or when used as cell donors for transplantation.

DOI: https://doi.org/10.7554/eLife.41258.005

The following figure supplements are available for figure 2:

**Figure supplement 1.** Label induction and progression in 1x Tamoxifen treated adult *Fgd5^CreERT2/+; Rosa26^lsl-Tomato/+* mice.
DOI: https://doi.org/10.7554/eLife.41258.006

**Figure supplement 2.** In vitro differentiation potential of LSK HSPCs.

*Figure 2 continued on next page*

*Figure 2 continued*

DOI: https://doi.org/10.7554/eLife.41258.007

Of the distinct progenitor fractions within the Lin⁻kit⁺ fraction (*Figure 1—figure supplement 1A*), megakaryocyte progenitors (MkP) acquired label with the fastest kinetics, reaching label equilibrium with HSCs after 32 weeks. Other myeloerythroid progenitors, including pre-megakaryocytic/erythroid (preMeg/E), pre-colony forming unit-erythroid (pre CFU-E) and pre-granulocyte-macrophage (preGM) progenitors acquired Tomato label with very similar kinetics despite their distinct lineage affiliations, although they never quite reached an equilibrium with HSCs throughout the course of the experiments (*Figure 2B* and *Figure 2—figure supplement 1*).

Mature effector cells represent the terminal progeny of HSCs. We observed distinct generation kinetics for different lineages (*Figure 2B*). First, we made the general observation that myeloerythroid cells acquired label more rapidly than lymphoid cells. Among the myeloid subsets, platelets acquired Tomato label with the fastest kinetics, followed by granulocytes and erythrocytes. Among lymphoid cell types, NK cells displayed faster labeling kinetics followed by B cells. T cells showed the slowest labeling kinetics among lymphoid cells and CD4⁺ T cells acquired label faster than CD8⁺ T cells (*Figure 2B*). Because the frequency of Tomato⁺ cells increased over time in all evaluated lineages, this data demonstrate a continuous contribution of HSCs to all hematopoietic lineages.

While multiple studies have defined populations of hematopoietic progenitors that associate with distinct developmental and/or stages of differentiation (*Bryder et al., 2006*), it is unknown whether such described progenitors are obligatory intermediates and/or their quantitative association relative to their anticipated mature offspring. Therefore, we interrogated the relationships between the rates of (re)generation of candidate committed myeloerythroid progenitors to those of their proposed mature cell lineage. At the earliest time points evaluated, we observed for all evaluated fractions a higher label in their corresponding progenitors (*Figure 2D*). However, this was resolved during the course of the experiments and reached similar equilibrium ratios for all evaluated lineages, although the erythroid lineage displayed somewhat slower kinetics (*Figure 2D*). Collectively, these experiments are in line with the view that progenitor generation precedes the generation of mature cells and that previously proposed progenitors appears to be, at least for the most part, obligatory intermediates.

Hematopoiesis after transplantation of HSCs is fundamentally different from unperturbed hematopoiesis (*Busch et al., 2015*; *Sun et al., 2014*). However, to what extent the pre-conditioning regimen and co-transplantation of mature cells and progenitors influence on hematopoiesis from HSCs is less established. Therefore, we next transplanted wild type recipient mice on continuous Tamoxifen diet with purified *Fgd5*^CreERT2/+^; *Rosa26*^lsl-Tomato/+^ HSCs or WBM cells. Here, recipient mice were pre-conditioned by either lethal irradiation or antibody mediated CD45-depletion (*Palchaudhuri et al., 2016*). Due to the HSC specificity of the model, this approach allowed us to monitor the kinetics of the HSC contribution to all lineages after transplantation and compare it to the HSC contribution in steady state (*Figure 2E*). Compared to steady state, label progression in transplanted mice were faster (*Figure 2E*). When label progression kinetics was compared between HSC and WBM transplanted animals, HSC transplantation resulted in faster label progression, especially into the B cell lineage (*Figure 2E*). This likely reflects a significant contribution to the regeneration of the B cell lineage by co-transplanted long-lived B-lineage progenitors and mature cells after WBM transplantation. Comparison of label progression after WBM transplantation into irradiated or non-irradiated/antibody-mediated conditioned recipient mice revealed similar label progression kinetics into most mature lineages, with the exception of platelets that displayed a faster label progression in irradiated mice. This suggests that progenitors for platelets are more effectively ablated by irradiation than antibody-mediated pre-conditioning.

## Fgd5-mediated HSC lineage tracing corroborates the fetal/juvenile origin of Langerhans cells, B1a B cells and brain microglial cells

While a labeling regimen of one Tamoxifen injection allows for accurate kinetic evaluations (*Figure 2A–B,D*), this experimental strategy labels only a fraction of HSCs (*Figure 1E* and *Figure 2A*) and thus necessitates correlation of label in HSCs to other evaluated cell subsets

(*Busch et al., 2015*) (*Figure 2B*). If the original HSC label is low, this might as a consequence not allow for evaluation of the activity of the entire pool of HSCs.

To explore whether we could label the HSC pool more extensively, *Fgd5$^{CreERT2/+}$; Rosa26$^{lsl-Tomato/+}$* mice were fed Tamoxifen containing food pellets for 16 weeks. To rule out adverse effects of prolonged Tamoxifen treatment on HSC proliferation, these experiments were preceded by a control label retaining experiment using *Col1a1$^{tetO-H2B-mCherry/tetO-H2B-mCherry}$; ROSA26$^{rtTA/rtT}$* mice (*Säwén et al., 2016*). Following H2B-mCherry induction with Doxycycline, mice were chased for 5 weeks in the presence or absence of Tamoxifen. Prolonged Tamoxifen treatment did not induce any additional proliferation within the HSC compartment, while more differentiated progenitors had readily proliferated in both settings (*Figure 3A*).

The 16 weeks labeling period was followed by an extensive (up to 41 weeks) chase period, during which mice received normal chow (*Figure 3B*). This labeling strategy resulted in labeling of virtually all candidate HSCs (*Figure 3C*). The blood of labeled mice was analyzed regularly to determine the fraction of Tomato$^+$ cells in PB cell subsets (*Figure 1—figure supplement 1B*). Similar to after 1x Tamoxifen labeling, we observed robust label progression into all PB cell subsets, with similar kinetics in between different lineages (*Figure 3B*). However, a more complete HSC labeling resulted in a somewhat faster and more robust label progression into all PB cell subsets compared to 1x Tamoxifen labeling (*Figure 2E*). This was most evident for the lymphoid lineages, where the majority of PB cells had been generated from HSCs at the experiment end point upon prolonged Tamoxifen administration, whereas the ratio of labeled lymphocytes vs. labeled HSCs was low (>0,5) even after 83 weeks of chase in 1x Tamoxifen labeled mice (*Figure 2B*, *Figure 2E* and *Figure 3B*).

From endpoint mice in which the pool of HSCs was almost completely labeled (*Figure 3C*), we next interrogated the skin epidermis for Tomato$^+$ contribution to granulocytes and Langerhans cells. Granulocytes were almost completely Tomato positive, while Langerhans cells were devoid of label (*Figure 3D*, lower left), in line with the fetal origin and self-maintenance of the latter cells (*Collin and Milne, 2016*). Next, we interrogated Tomato expression in Vγ3δ$^+$ T cells, an established fetal derived T cell subset (*Havran and Allison, 1990*). To our surprise, this revealed robust Tomato labeling of Vγ3δ$^+$ T cells (*Figure 3D*, lower left). However, closer examination revealed high expression of ZsGreen in these cells (*Figure 3B*, lower middle). Therefore, rather than establishing adult contribution to this lineage, these experiments established Fgd5-CreERT2 transgene expression in Vγ3δ$^+$ T cells.

B1a B cells represent an invariant subtype of B cells with a fetal origin that is primarily located in the peritoneum (*Hayakawa et al., 1985*), where they co-exists with more traditional B1b and B2 B cells in adult mice. While less than 10% of B1a B cells displayed Tomato label, around 50% of B1b B cells and the vast majority of B2 B cells (*Figure 3D* right) were Tomato$^+$ (comparable to levels in PB). This is in line with a more strict fetal/postnatal origin of B1a B cells, the ontogenically mixed origin of B1b B cells (*Kantor et al., 1995*) and an adult HSC origin of most B2 B cells.

Finally, we investigated adult HSC contribution to microglial cells of the brain, a subset of central nervous system myeloid cells that has been proposed to arise entirely from embryonic precursor cells (*Alliot et al., 1999*). Evaluations by confocal microscopy of the brain parenchyma revealed no detectable Tomato expression in IBA-1$^+$ microglia (*Figure 3E*), while Fgd5 expressing endothelial cells (*Cheng et al., 2012*; *Gazit et al., 2014*) displayed abundant Tomato expression (*Figure 3E* middle).

## Native hematopoiesis from HSCs declines with aging

We next set out to investigate how chronological aging influence on HSC contribution to hematopoiesis. To achieve rapid and robust labeling of HSCs, we labeled juvenile and aged *Fgd5$^{CreERT2/+}$; Rosa26$^{lsl-Tomato/+}$* mice by injecting Tamoxifen for five consecutive days (5x). Labeling was followed by evaluation of the fraction of Tomato positive cells in HSC and MPP fractions of the BM LSK compartment one day after the last Tamoxifen injection.

In aged mice, the initial labeling was highly specific to HSCs, with only low levels of labeling in MPP$^2$ cells. In sharp contrast, a larger fraction of LSKCD150$^-$CD48$^-$ MPPs were labeled in juvenile mice (*Figure 4A*). Next, we correlated how increasing age influences on the HSC generation of other LSK/MPP subsets. *Fgd5$^{CreERT2/+}$; Rosa26$^{lsl-Tomato/+}$* mice between 6 and 96 weeks of age were labeled using a 5x Tamoxifen injection scheme, before evaluation of Tomato label in HSCs/MPP$^{2-4}$21 days later. Ratios of the fraction of labeled MPPs vs. labeled HSCs in corresponding mice was

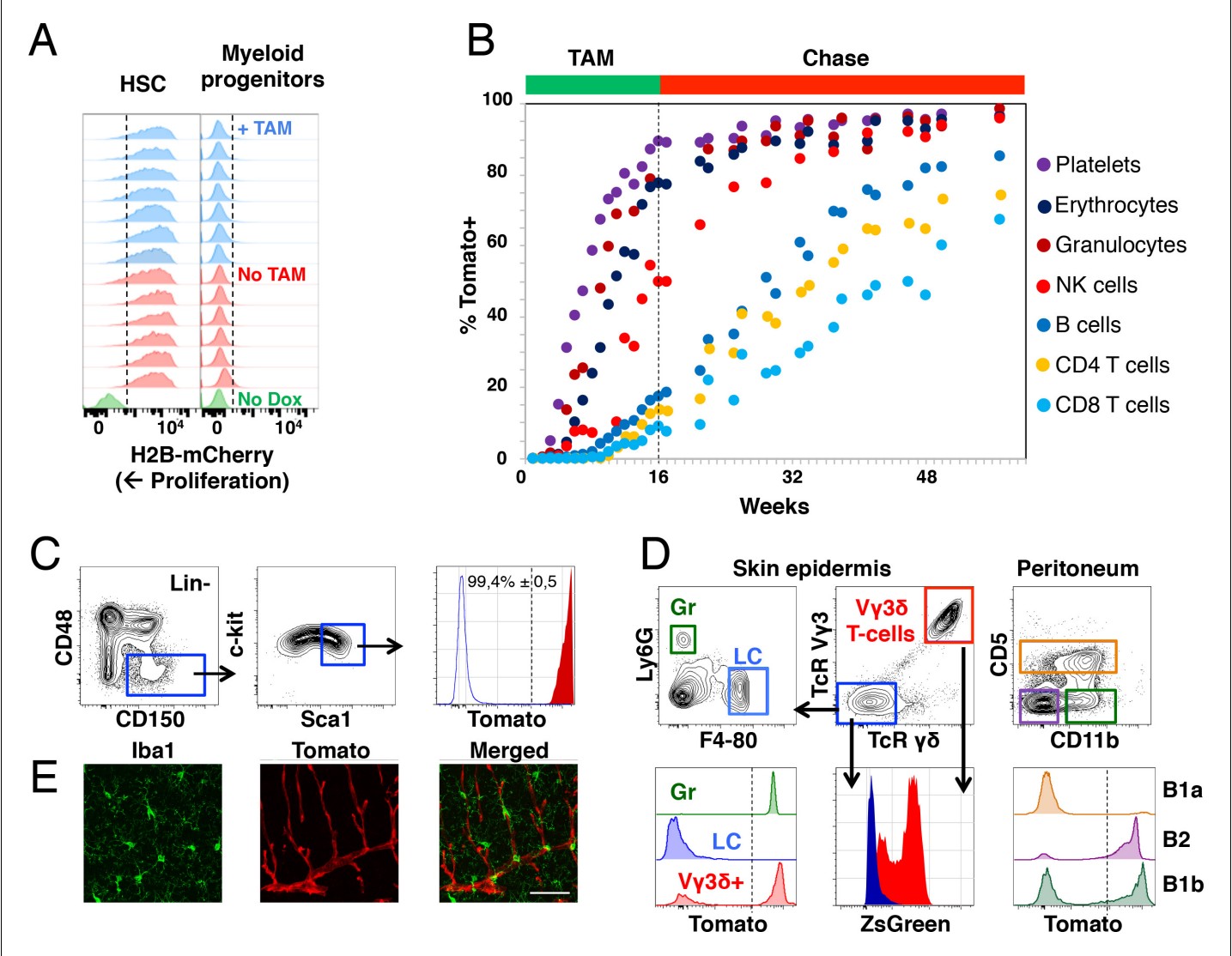

**Figure 3.** Fgd5-mediated lineage tracing after complete HSC labeling reveals limited adult HSCs contribution to tissue-resident immune cell subsets. (A) H2B-mCherry label retention in HSCs and myeloid progenitors after 5 weeks of chase in mice continuously fed Tamoxifen (TAM) containing food (blue histograms, n = 7) or normal food (red histograms, n = 6) during the chase period. Green histograms depict an unlabeled control. Dashed lines indicate the boundary for H2B-mCherry positivity. Lineage negative c-kit⁺Sca1⁻ cells are denoted as myeloid progenitors. (B) Cohorts of *Fgd5^CreERT2/+*; *Rosa26^lsl-Tomato/+* mice were continuously fed Tamoxifen food for 16 weeks (TAM phase) and thereafter normal chow during a chase phase of 32 weeks (n = 5) or 41 weeks (n = 4). During the TAM and chase phases, PB was regularly analyzed for Tomato label in the indicated subsets. Data points indicate the average % of Tomato label in each indicated subset. (C–E) Endpoint analysis, after 41 weeks chase, in mice from *Figure 3B* (n = 3). (C) Representative FACS plots showing the gating strategy to identify HSCs in lineage negative BM cells (left, middle) and depiction of the Tomato label in HSCs (right, red histogram) compared to an unlabeled control (blue histogram). Dashed line indicates the boundary for Tomato positivity, arrows indicate the gating strategy. (D) Representative FACS plots of cells isolated from skin tissue (epidermis) or the peritoneal cavity. Histograms show Tomato label (bottom left and right) or ZsGreen label (bottom middle) in the indicated subsets. Dashed line indicate the boundary for Tomato positivity, arrows indicate the gating hierarchy, gates and corresponding histograms are color matched. (E) Representative confocal images of the brain. (Left) IBA1 positive cells (green) are not labeled with Tomato (red, middle) while blood vessels are labeled with Tomato (middle). (Right) Merged images display both IBA1 staining and Tomato label. Scale bar = 50 μm. Mice in B-E were 5–6 weeks old at the start of Tamoxifen administration.
DOI: https://doi.org/10.7554/eLife.41258.008

calculated and plotted against mouse age at labeling (*Figure 4B*). This established that label progression into all MPP subsets in aged mice was substantially lower when compared to young adult mice and further revealed that the HSC contribution to MPPs and MPP³/⁴s gradually declines with age towards very little replenishment of in particular MPP³/⁴ in very old age (*Figure 4B*).

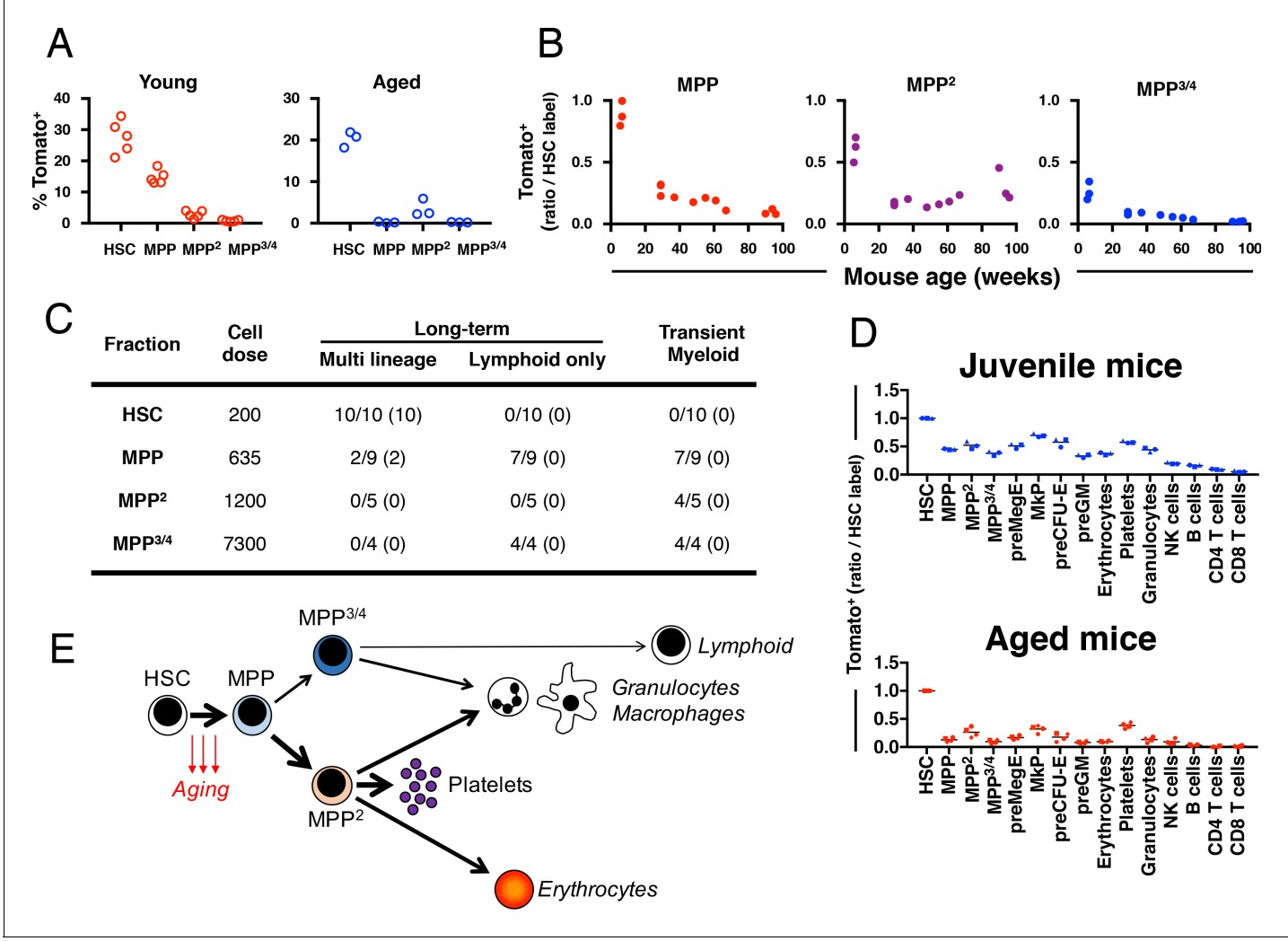

**Figure 4.** Fgd5 mediate lineage tracing reveals gradually declining HSC contribution to hematopoiesis with age. (**A**) Cohorts of aged (11–12 months; n = 3) or young (23–25 days; n = 5,) $Fgd5^{CreERT2/+}$; $Rosa26^{lsl-Tomato/+}$ mice were injected with Tamoxifen 5 times on consecutive days and analyzed for Tomato label in LSK-HSPCs on day 6. Bars indicate median %. (**B**) $Fgd5^{CreERT2/+}$; $Rosa26^{lsl-Tomato/+}$ mice were injected with Tamoxifen for 5 consecutive days and analyzed for Tomato label in LSK-HSPCs 21 days later. The % of Tomato$^+$ cells in MPP$^{2-4}$ was divided by the % of Tomato$^+$ HSCs in corresponding mice to determine the ratio of Tomato$^+$ cells relative to HSCs. Ratios are plotted according to the age of mice at the time of Tamoxifen injection. (**C**) Indicated LSK subsets were isolated from $Fgd5^{CreERT2/+}$; $Rosa26^{lsl-Tomato/+}$ mice and transplanted into WT mice on continuous Tamoxifen. Multilineage hematopoiesis and Tomato contribution was assed in PB 4, 8, 12 and 16 weeks after transplantation. Values in parenthesis indicate the number of mice with Tomato$^+$ offspring. (**D**) Young (29 days; n = 3) and old (16 months; n = 4) $Fgd5^{CreERT2/+}$; $Rosa26^{lsl-Tomato/+}$ mice were injected once with Tamoxifen and analyzed for Tomato label 18 weeks later. The % of Tomato$^+$ cells in each indicated cell type was divided by the % of Tomato$^+$ HSCs in corresponding mice to determine the ratio of Tomato$^+$ cells relative to HSCs. (**E**) Model depicting HSC contribution to native hematopoiesis. Arrow weights and arrow lengths indicate the magnitude and the kinetics of HSC contribution to the indicated cell type respectively. The declining HSC contribution to hematopoiesis with age can be traced to a reduced differentiation of HSCs to MPPs (red arrows).

DOI: https://doi.org/10.7554/eLife.41258.009

The following figure supplement is available for figure 4:

**Figure supplement 1.** Aging associates with reduced multilineage HSC contribution.
DOI: https://doi.org/10.7554/eLife.41258.010

To evaluate the functional potential of initially Tomato-labeled MPPs, we FACS sorted HSCs and different MPP subsets isolated from non-labeled $Fgd5^{CreERT2/+}$; $Rosa26^{lsl-Tomato/+}$ mice and transplanted cells into lethally irradiated wild type mice. Recipient mice were provided Tamoxifen containing food pellets throughout the experiment. We analyzed donor chimerism and Tomato labeled PB cells after repetitive PB blood sampling. As expected, we did not observe any multilineage long-

term reconstitution or Tomato$^+$ cells in MPP$^2$/MPP$^{3/4}$ transplanted mice (*Figure 4C*). 10 out of 10 HSC transplanted mice were multi lineage reconstituted at 16 weeks post transplantation with high levels of Tomato$^+$ donor cells in all evaluated lineages (*Figure 4C*). More surprisingly 2 out of 9 MPP transplanted mice displayed donor reconstitution levels > 1% in all lineages 16 weeks after transplantation (*Figure 4C*). This long-term multilineage reconstitution potential from MPPs was accompanied with robust Tomato labeling among donor cells and revealed that Tomato labeled phenotypic MPPs perform as bona fide HSCs after transplantation (*Figure 4C*). By contrast, mice that received MPP cells and displayed only transient myeloid reconstitution never displayed any Tomato$^+$ cells. This demonstrates, in young mice, the presence of a minor CD150$^-$ HSC activity that appears exclusively coupled to Fgd5 expression.

Finally, we were interested in determining whether the age-related decrease in HSC derived MPPs might influence on the generation kinetics of lineage-restricted progenitors and mature blood cells. For this, we labeled juvenile (29 days) and aged (87–89 weeks) *Fgd5$^{CreERT2/+}$; Rosa26$^{lsl-Tomato/+}$* mice with 1x Tamoxifen, which was followed by evaluation of label progression after 18 weeks (*Figure 4D* and *Figure 4—figure supplement 1*). While the frequency of Tomato labeled HSCs was similar among the two groups (*Figure 4—figure supplement 1*), all progeny exhibited reduced frequencies of Tomato labeled cells in aged mice, demonstrating a strikingly reduced multilineage differentiation capacity of HSCs as a consequence of age.

## Discussion

In this work, we explored the cellular contribution from HSCs using a HSC-specific lineage tracing approach. Our work revealed robust HSC contribution to adult multilineage hematopoiesis and, while not focused at studying fetal hematopoiesis, support the previously reported fetal/juvenile origins of several other specific hematopoietic subsets. We observed the fastest label progression into the platelet lineage, which might be related to the recent observations that at least a subset of HSCs appears platelet-biased (*Sanjuan-Pla et al., 2013*; *Shin et al., 2014*). Thereafter, erythrocytes and granulocytes acquired label with similar kinetics, although Tomato$^+$ erythrocytes emerged in the peripheral blood somewhat later than granulocytes. This likely reflects the slower turnover of mature erythrocytes compared to other myeloid cells. On the other hand, among granulocytes, we did not find any noticeable differences when evaluating HSC contribution to the neutrophil or eosinophil lineages (data not shown), despite the seemingly distinct transcriptional underpinnings of these lineages (*Drissen et al., 2016*).

Among lymphocytes, the HSC contribution was most rapid and robust to the NK cell lineage. At present, we can say little on whether this reflects an early and distinct progenitor intermediate for NK cells (*Wu et al., 2014*) or whether NK cells are, at least partially, regenerated through myeloid progenitors (*Chen et al., 2015*; *Grzywacz et al., 2011*). While the extremely slow HSC generation of adaptive immune components (and T cells in particular) have implications for our understanding of HSCs, by proposing that adult HSCs to a large extent present as myeloid-biased in an unperturbed scenario, our results also suggest the necessity of very harsh conditioning to achieve an 'immunological reboot' (the generation of naïve lymphocytes) in certain autoimmune situations (*Atkins et al., 2016*).

According to current models of hematopoiesis, lineage committed progenitors reside developmentally in between HSCs and their mature progeny, and much work has been aimed at detailing these stages (*Adolfsson et al., 2005*; *Akashi et al., 2000*; *Arinobu et al., 2007*; *Oguro et al., 2013*; *Pietras et al., 2015*; *Pronk et al., 2007*). What has remained more unknown is the relationships of these defined progenitors not only to HSCs and their mature offspring, but also whether they are obligatory. In our work, we could demonstrate that the generation rates of each evaluated myeloid progenitor subset correlated highly to their corresponding mature offspring, although the generation kinetics varied depending on lineage. Thus, although some challenges have recently been raised on how lineage commitment occur from HSCs based on inferences from large-scale single-cell RNA sequencing experiments (*Paul et al., 2015*; *Velten et al., 2017*), our data support the more conventional view that the generation of mature myeloid cells is preceded by the generation of obligatory lineage-committed intermediates.

Investigations of immature progenitors revealed that the most rapid label progression associated with LSKCD150$^-$CD48$^-$ MPPs. MPPs share many defining properties of HSCs, including multilineage

differentiation potential and very low proliferation rates in steady state (*Säwén et al., 2016*). The distinction of MPPs from HSCs is mainly thought to result from pronounced differences in self-renewal; a property so far entirely evaluated by transplantation. Intriguingly, our work revealed that LSKCD150⁻CD48⁻ MPPs displayed more rapid label kinetics in young mice, which was gradually declining with advancing age. At the same time, we could in agreement with previous studies (*Weksberg et al., 2008*) also demonstrate a minor multilineage HSC activity in this compartment, which correlated exclusively to Fgd5 expression/Tomato labeling. Together with an overall decline in multilineage HSC contribution of aged mice, these results strongly propose a model in which aging associates with reduced/compromised HSC differentiation, which in combination with the well-established expansion of HSCs with age (*Morrison et al., 1996*; *Rossi et al., 2005*; *Sudo et al., 2000*) appears to represent a physiologically relevant compensatory mechanisms to sustain multilineage hematopoiesis from HSCs (*Figure 4E*).

Compared to MPPs, MPP$^{3/4}$ are perhaps easier to approach given their restrictions in lineage potential (lack of Meg/E potential) (*Adolfsson et al., 2005*; *Arinobu et al., 2007*; *Pietras et al., 2015*; *Pronk et al., 2007*). We found MPP$^{3/4}$ to be regenerated from HSCs with slow kinetics compared to other downstream myeloid progenitor cells, but also to MPP$^2$ cells, that we in agreement with other studies (*Pietras et al., 2015*) find 'primed' towards Meg/E development. Intriguingly, our data proposes a significant self-renewal activity of at least a subset of MPP$^{3/4}$, with the demonstration that this fraction never reached label equilibrium with HSCs in any evaluated experimental setting. This might be particularly relevant in the setting of age, a situation in which HSC was found to generate MPP$^{3/4}$ very inefficiently (*Figure 4E*).

While limited, a few groups have recently approached HSC contribution to native hematopoiesis. Evaluations of hematopoiesis using transposon mobilization led to the conclusion that HSCs are not major contributors to adult hematopoiesis (*Sun et al., 2014*). To some degree, this conclusion was later corroborated by CreER-mediated labeling of a minor fraction of the adult HSC pool using a Tie2-based CreER driver (*Busch et al., 2015*). Limited HSC contribution to adult hematopoiesis is in sharp contrast to the results we present here and to results from another recent study (*Sawai et al., 2016*). Our studies would propose that absence of a HSC specific driver, as in the work from Sun et al., makes interpretations of HSC contribution very complicated, not the least for the lymphoid lineages, while the labeling of only a minor fraction of HSCs, as in the work from Busch et al., might select for a subset of HSCs with a rather distinct functional behavior.

In summary, we conclude that although the study of native hematopoiesis highlights fundamental differences, with in particular slower regeneration times from HSCs to those seen after transplantation, they regardless corroborate decades of research derived from transplantation experiments in which HSCs has been proposed to continuously contribute to hematopoiesis.

# Materials and methods

## Key resources table

| Reagent type (species) or resource | Designation | Source or reference | Identifiers | Additional information |
|---|---|---|---|---|
| Strain, strain background (M. musculus) | *Fgd5-CreERT2* | PMID:24958848 | RRID:IMSR_JAX:027789 | |
| Strain, strain background (M. musculus) | *Rosa26-rtTA; Col1a1-tetO-H2B-mCherry* | PMID:17554301 | RRID:IMSR_JAX:014602 | |
| Strain, strain background (M. musculus) | *Rosa26-Lox-Stop-Lox-Tomato* | PMID:20023653 | RRID:IMSR_JAX:007905 | |
| Antibody | B220 PECy5  Clone: RA3-6B2 | Biolegend | RRID:AB_312994 | (1:400) |

*Continued on next page*

*Continued*

| Reagent type (species) or resource | Designation | Source or reference | Identifiers | Additional information |
|---|---|---|---|---|
| Antibody | B220 biotin<br>Clone: RA3-6B2 | Biolegend | RRID:AB_312989 | (1:200) |
| Antibody | B220 APC<br>Clone: RA3-6B2 | Biolegend | RRID:AB_312997 | (1:400) |
| Antibody | CD105 PECy7<br>Clone: MJ7/18 | Biolegend | RRID:AB_1027700 | (1:200) |
| Antibody | CD115 BV605<br>Clone: CSF-1R | Biolegend | RRID:AB_2562760 | (1:200) |
| Antibody | CD11b PECy5<br>Clone: M1/70 | Biolegend | RRID:AB_312793 | (1:400) |
| Antibody | CD11b biotin<br>Clone: M1/70 | Biolegend | RRID:AB_312787 | (1:200) |
| Antibody | CD11b APC<br>Clone: M1/70 | Biolegend | RRID:AB_312795 | (1:800) |
| Antibody | CD11b APC-Cy7<br>Clone: M1/70 | Biolegend | RRID:AB_830641 | (1:200) |
| Antibody | CD11c BV570<br>Clone: N418 | Biolegend | RRID:AB_10900261 | (1:200) |
| Antibody | CD150 APC<br>Clone: TC15-12F12.2 | Biolegend | RRID:AB_493460 | (1:400) |
| Antibody | CD150 PE<br>Clone: TC15-12F12.2 | Biolegend | RRID:AB_313683 | (1:200) |
| Antibody | CD16/32 AL700<br>Clone: 93 | eBioscience | RRID:AB_493994 | (1:100) |
| Antibody | CD19 PECy7<br>Clone: 1D3 | eBioscience | RRID:AB_657663 | (1:200) |
| Antibody | CD19 BV-786<br>Clone: 1D3 | BD-Horizon | RRID:AB_2738141 | (1:200) |
| Antibody | CD25 APC<br>Clone: PC 61.5 | eBioscience | RRID:AB_469366 | (1:100) |
| Antibody | CD31 PerCpCy5.5<br>Clone: MEC13.3 | Biolegend | RRID:AB_2566761 | (1:200) |
| Antibody | CD3e PECy5<br>Clone: 145–2 C11 | Biolegend | RRID:AB_312675 | (1:400) |
| Antibody | CD3e biotin<br>Clone: 17A2 | Biolegend | RRID:AB_2563947 | (1:200) |
| Antibody | CD3 AL700<br>Clone: 17A2 | Biolegend | RRID:AB_493697 | (1:200) |

*Continued on next page*

*Continued*

| Reagent type (species) or resource | Designation | Source or reference | Identifiers | Additional information |
|---|---|---|---|---|
| Antibody | CD4 BV711 | Biolegend | RRID:AB_2562607 | (1:200) |
| | Clone: RM4-5 | | | |
| Antibody | CD4 APC eFl780 | eBioscience | RRID:AB_1272219 | (1:200) |
| | Clone: RM4-5 | | | |
| Antibody | CD41 PerCP-eFl710 | eBioscience | RRID:AB_10855042 | (1:200) |
| | Clone: MWReg30 | | | |
| Antibody | CD45.1 AL700 | Biolegend | RRID:AB_493733 | (1:200) |
| | Clone: A20 | | | |
| Antibody | CD45.2 PECy7 | Biolegend | RRID:AB_1186098 | (1:200) |
| | Clone: 104 | | | |
| Antibody | CD45.2 PE/Dazzle | Biolegend | RRID:AB_2564177 | (1:200) |
| | Clone: 104 | | | |
| Antibody | CD45.2 BV785 | Sony | RRID:AB_2562604 | (1:200) |
| | Clone: 104 | | | |
| Antibody | CD45.2 biotin | Biolegend | RRID:AB_313441 | (1:200) |
| | Clone: 104 | | | |
| Antibody | CD48 PECy7 | Biolegend | RRID:AB_2075049 | (1:200) |
| | Clone: HM48-1 | | | |
| Antibody | CD48 AL700 | Biolegend | RRID:AB_10612755 | (1:200) |
| | Clone: HM48-1 | | | |
| Antibody | CD5 BV-421 | BD-Horizon | RRID:AB_2737758 | (1:200) |
| | Clone: 53–7.3 | | | |
| Antibody | CD8 PerCpCy5.5 | Sony | RRID:AB_2075239 | (1:200) |
| | Clone: 53–6.7 | | | |
| Antibody | c-kit APCeFl780 | eBioscience | RRID:AB_1272177 | (1:200) |
| | Clone: 2B8 | | | |
| Antibody | c-kit APC | Biolegend | RRID:AB_313221 | (1:100) |
| | Clone: 2B8 | | | |
| Antibody | F4/80 BV421 | Biolegend | RRID:AB_11203717 | (1:200) |
| | Clone: BM8 | | | |
| Antibody | Flt3 biotin | eBioscience | RRID:AB_466600 | (1:200) |
| | Clone: AZF10 | | | |
| Antibody | Gr1 PECy5 | Biolegend | RRID:AB_313375 | (1:400) |
| | Clone: RB6-8C5 | | | |
| Antibody | Gr1 FITC | BD PH | RRID:AB_394643 | (1:400) |
| | Clone: RB6-8C5 | | | |
| Antibody | Gr1 biotin | Biolegend | RRID:AB_313369 | (1:200) |
| | Clone: RB6-8C5 | | | |

*Continued on next page*

*Continued*

| Reagent type (species) or resource | Designation | Source or reference | Identifiers | Additional information |
|---|---|---|---|---|
| Antibody | Gr1 BV711<br>Clone: RB6-8C5 | Sony | RRID:AB_2562549 | (1:200) |
| Antibody | IL7Ra BV510<br>Clone: A7R34 | Sony | RRID:AB_2564576 | (1:200) |
| Antibody | Ly6G APC/Fire750<br>Clone: 1A8 | Biolegend | RRID:AB_2616733 | (1:200) |
| Antibody | NK1.1 PECy5<br>Clone: PK136 | Biolegend | RRID:AB_493591 | (1:400) |
| Antibody | NK1.1 Pacific Blue<br>Clone: PK136 | Biolegend | RRID:AB_2132712 | (1:200) |
| Antibody | Sca1 Pacific Blue<br>Clone: E13-161.7 | Biolegend | RRID:AB_2143237 | (1:200) |
| Antibody | TcR Vγ3 APC<br>Clone: 536 | Biolegend | RRID:AB_10895900 | (1:200) |
| Antibody | TcR γ/δ BV605<br>Clone: GL3 | Biolegend | RRID:AB_2563356 | (1:200) |
| Antibody | Ter119 PECy5<br>Clone: Ter-119 | Biolegend | RRID:AB_313711 | (1:400) |
| Antibody | Ter119 biotin<br>Clone: Ter-119 | Biolegend | RRID:AB_313705 | (1:200) |
| Antibody | Ter119 PerCpCy5.5<br>Clone: Ter-119 | Biolegend | RRID:AB_893636 | (1:200) |
| Peptide, recombinant protein | Streptavidin BV605 | Biolegend | | (1:400) |
| Peptide, recombinant protein | Streptavidin-Saporin | PMID: 27272386 Advanced Targeting Systems | | |
| Chemical compound, drug | Tamoxifen | Sigma-Aldrich | I.p. 50 mg/kg | |
| Chemical compound, drug | Doxycycline | Ssniff Spezialdiäten | Food 2 g/kg | |
| Chemical compound, drug | Tamoxifen | Ssniff Spezialdiäten | Food 400 mg/kg Tamoxifen Citrate | |
| Software, algorithm | Flowjo | FlowJo (https://www.flowjo.com/solutions/flowjo) | RRID:SCR_008520 | |

*Continued on next page*

*Continued*

| Reagent type (species) or resource | Designation | Source or reference | Identifiers | Additional information |
|---|---|---|---|---|
| Software, algorithm | Microsoft Excel | Microsoft Excel (https://www.microsoft.com/en-gb/) | RRID: SCR_016137 | |
| Software, algorithm | Graphpad Prism | GraphPad Prism (https://graphpad.com) | RRID: SCR_002798 | |

## Mouse procedures

For inducible marking of HSCs in vivo, we crossed *Fgd5-2A-ZsGreen-CreERT2* mice (*Gazit et al., 2014*) (JAX 027789) to *Rosa26-LoxP-Stop-LoxP-Tomato* (*Madisen et al., 2010*) (JAX 007905) mice, resulting in *Fgd5$^{CreERT2/+}$; Rosa26$^{lsl-Tomato/+}$* mice. For simultaneous in vivo tracking of proliferation history and marking of HSCs, *Fgd5$^{CreERT2/+}$; Rosa26$^{lsl-Tomato/+}$* mice were crossed with *Col1a1$^{tetO-H2B-mCherry/tetO-H2B-mCherry}$; ROSA26$^{rtTA/rtTA}$* mice (JAX 014602) to generate *Fgd5$^{CreERT2/+}$; Rosa26$^{lsl-Tomato/rtTA}$; Col1a1$^{tetO-H2B-mCherry/+}$*. Such mice were administered doxycycline in food pellets (2 g/kg; Ssniff Spezialdiäten) for 2 weeks followed by 5 weeks of chase before HSC marking by a single i.p. Tamoxifen injection (50 mg/kg) and analysis 5 days later.

Tamoxifen was purchased from Sigma-Aldrich and suspended at 100 mg/ml in ethanol and mixed with sunflower oil to a concentration of 10 mg/ml. Tamoxifen was administered by intraperitoneal injections at 50 mg/kg body weight once (1x) or for 5 (5x) consecutive days. To acquire full/maximal HSC labeling, cohorts of mice were continuously fed Tamoxifen containing food pellets for 16 weeks. Mice on Tamoxifen food were regularly bled (sparsely; 1–2 drops) during the labeling period and during the chase period.

Transplanted recipient mice were subjected to lethal irradiation (950 rad) except CD45.1/2 mice that were CD45-depleted by intravenous injection of an immunotoxin (3 mg/kg) consisting of CD45.2-biotin (clone 104) and streptavidin-Saporin (Advanced Targeting Systems) in a 1:1 molar ratio 3 days prior to transplantation, as described (*Palchaudhuri et al., 2016*). All transplanted cells were isolated from CD45.2 *Fgd5$^{CreERT2/+}$; Rosa26$^{lsl-Tomato/+}$* mice. CD45-depleted recipients were transplanted with $10^7$ whole bone marrow (WBM) cells and irradiated mice were transplanted with $3 \times 10^6$ WBM cells (n = 7), 200 (n = 10) or 100 (n = 2) HSCs, 635 MPPs (n = 10), 1200 MPP$^2$s (n = 5) or 7300 MPP$^{3/4}$s (n = 4). Tomato$^+$ and Tomato$^-$ HSCs were isolated from *Fgd5$^{CreERT2/+}$; Rosa26$^{lsl-Tomato/+}$* mice injected with Tamoxifen 2 days before isolation and transplantation into congenic C57BL/6 (CD45.1$^+$) mice on normal chow. Before transplantations of HSCs or MPPs FACS sorted cells were mixed with $3 \times 10^5$ WBM competitor cells in 200 µl PBS supplemented with 2 mM EDTA and 2% FBS before injection. Where indicated, recipient mice were given Tamoxifen containing food pellets (400 mg/kg Tamoxifen Citrate, Ssniff Spezialdiäten) throughout the experiments. At the indicated time point, PB was collected from the tail vein for reconstitution analysis.

H2B-mCherry labeling in *Col1a1$^{tetO-H2B-mCherry/tetO-H2B-mCherry}$; ROSA26$^{rtTA/rtTA}$* mice was induced by administration of doxycycline (*Säwén et al., 2016*). Thereafter, mice were chased for 5 weeks while eating Tamoxifen containing food pellets or normal chow (No TAM), followed by FACS analysis to assess H2B-mCherry dilution in HSCs/progenitors.

All mice were maintained at the animal facilities at BMC at Lund University and all experiments were performed with consent from the Malmö/Lund animal ethics board, reference number M186-15.

## Immunophenotyping and FACS

Immunophenotyping by FACS was done as described (*Säwén et al., 2016*) (*Supplementary file 1*). For platelet and erythrocyte analysis, 1 µl of whole blood was taken to 300 µl PBS before FACS analysis. Cells were sorted and/or analyzed on a FACS Aria III cell sorter (Becton Dickinson) or on a LSRFortessa (Becton Dickinson).

### B1a, Langerhans cells and Vg3[+] T cell analysis

For isolation of peritoneal cells, peritoneal lavage was performed using 10 mL PBS. For isolation of skin epidermal cells, the flank of the mouse was shaved before excision of skin. The skin was incubated for 25 min at 37°C in a dissociation buffer (PBS containing 2.4 mg/ml of dispase (Roche) and 3% FCS) before separation of dermis from the epidermis. Pieces of epidermis were incubated for 30 min at 37°C in digestion buffer (PBS supplemented with 1 mg/ml collagenase IV [Sigma-Aldrich], 100 U/ml DNase I [Sigma], 2.4 mg/ml dispase [Roche] and 3% FBS) and thereafter filtered and stained against indicated markers. Before analysis, cells were incubated with Propidium Iodide (Invitrogen) to exclude dead cells.

### Immunohistochemistry

Mice were deeply anaesthetized with an overdose of pentobarbital and transcardially perfused with cold saline. Brains were post-fixed for 48 hr in 4% paraformaldehyde (PFA) and incubated in 20% sucrose for 24 hr before being cut in 30 μm thick coronal sections on a microtome. Sections were incubated in blocking solution (5% normal serum and 0.25% Triton X-100 in 0.1 M potassium-phosphate buffered solution) for one hour and subsequently overnight at 4°C with the primary antibody (Iba1 1:1000 Wako). Fluorophore-conjugated secondary antibody (Molecular Probes or Jackson Laboratories) was diluted in blocking solution and applied for 2 hr at room temperature. Tomato label could be detected without any staining. Nuclei were stained with Hoechst (Molecular Probes) for 10 min and sections were mounted with Dabco mounting medium. Images were obtained using confocal microscopy (Zeiss, Germany).

### Cell culture

Single LSKCD150[+]CD48[+] cells, MPP[3/4]s and HSCs were sorted into Terasaki wells containing 20 μl of media (OptiMEM supplemented with 10% FCS, 1:1000 Gentamicin (Invitrogen), 1:100 GlutaMAX (Invitrogen) and 1:500 β-mercaptoethanol (Invitrogen) supplemented with cytokines (mSCF (Peprotech) 100 ng/ml, TPO (Peprotech) 10 ng/ml, IL-3 (Peprotech) 5 ng/ml, EPO (Janssen) 5units/ml, human G-CSF (Amgen) 10 ng/ml). After 6 days of culture at 37°C, wells were scored and evaluated for the presence of megakaryocytes and erythroid cells by visual inspection in microscope.

### Gene expression analyses

qRT-PCR analyses using the Fluidigm Biomark HD Platform was done as described (*Säwén et al., 2016*) (*Supplementary file 2*). PCA on gene expression data from all reference populations was performed using Clustvis, (http://biit.cs.ut.ee/clustvis/).

Single-cell RNA seq libraries were generated using a Chromium system (10x Genomics) according to the manufacturer's instructions. Two consecutive sequencing runs were performed to achieve enough sequencing depth and data was combined and further analyzed using the Cell Ranger[TM] pipeline (10x Genomics). The accession number for the single-cell RNA sequencing data reported in this paper is GSE122473.

### Statistical analysis

Data were analyzed using Microsoft Excel (Microsoft) and Graphpad Prism (GraphPad Software). All FACS analyses were performed using Flowjo software (TreeStar).

## Acknowledgments

We thank Gerd Sten for expert technical support and Martin Wahlestedt for assistance with manuscript editing. The work was aided by the access to the StemTherapy funded core facilities in Bioinformatics and flow cytometry. This work was supported by grants from the Tobias Foundation, the Swedish Cancer Society, the Swedish Medical Research Council, the Knut and Alice Wallenberg foundation and an ERC Consolidator grant (615068) to DB.

## Additional information

### Funding

| Funder | Grant reference number | Author |
|--------|------------------------|--------|
| Cancerfonden | | David Bryder |
| The Tobias foundation | Tobias Prize | David Bryder |
| Vetenskapsrådet | | David Bryder |
| Knut och Alice Wallenbergs Stiftelse | | David Bryder |
| European Research Council | ERC Consolidator grant (615068) | David Bryder |

The funders had no role in study design, data collection and interpretation, or the decision to submit the work for publication.

### Author contributions

Petter Säwen, Conceptualization, Formal analysis, Validation, Investigation, Visualization, Methodology, Writing—original draft, Project administration, Writing—review and editing; Mohamed Eldeeb, Investigation; Eva Erlandsson, Trine A Kristiansen, Zaal Kokaia, Joan Yuan, Methodology; Cecilia Laterza, Investigation, Methodology; Göran Karlsson, Formal analysis, Methodology; Shamit Soneji, Data curation, Formal analysis; Pankaj K Mandal, Derrick J Rossi, Resources; David Bryder, Conceptualization, Resources, Formal analysis, Supervision, Funding acquisition, Visualization, Writing—original draft, Project administration, Writing—review and editing

### Author ORCIDs

Petter Säwen (iD) http://orcid.org/0000-0001-6854-1434
Zaal Kokaia (iD) http://orcid.org/0000-0003-2296-2449
David Bryder (iD) http://orcid.org/0000-0002-8761-4237

### Ethics

Animal experimentation: All mice were maintained at the animal facilities at BMC at Lund University, and were performed with consent from a local ethical committee reference number M186-15

### Decision letter and Author response

Decision letter https://doi.org/10.7554/eLife.41258.017
Author response https://doi.org/10.7554/eLife.41258.018

## Additional files

### Supplementary files

• Supplementary file 1. Definitions of cells used throughout study.
DOI: https://doi.org/10.7554/eLife.41258.011

• Supplementary file 2. List of analyzed genes in mutiplexed qRT-PCR gene expression analyses (*Figure 2C*).
DOI: https://doi.org/10.7554/eLife.41258.012

• Transparent reporting form
DOI: https://doi.org/10.7554/eLife.41258.013

### Data availability

Sequencing data have been deposited in GEO under accession code GSE122473.

The following dataset was generated:

**Database and**

| Author(s) | Year | Dataset title | Dataset URL | Identifier |
|---|---|---|---|---|
| Säwén P, Erlands-son E, Soneji S, Bryder D | 2018 | HSCs contribute actively to native multilineage hematopoiesis but with reduced differentiation capacity upon aging | https://www.ncbi.nlm.nih.gov/geo/query/acc.cgi?acc=GSE122473 | Gene Expression Omnibus, GSE122473 |

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
