## [Decision Letter]

Thank you for submitting your article "HSCs contribute actively to native multilineage hematopoiesis but with reduced differentiation capacity upon aging" for consideration by *eLife*. Your article has been reviewed by two peer reviewers, one of whom is a member of our Board of Reviewing Editors, and the evaluation has been overseen by Sean Morrison as the Senior Editor. The reviewers have opted to remain anonymous.

The reviewers have discussed the reviews with one another and the Reviewing Editor has drafted this decision to help you prepare a revised submission.

The manuscript by Säwén et al. uses an HSC-specific Fgd5 reporter mouse strain to investigate steady state mouse hematopoiesis. Using this method, the authors demonstrate that HSCs continuously give rise to multilineage hematopoiesis, with kinetics increasing from platelets, to erythrocytes, granulocytes, NK cells, B cells, and T cells. They also provide evidence that these mature cells are derived from intermediate progenitor cells that are the progeny of HSCs. Moreover, they corroborate recent studies demonstrating that a subset of hematopoietic cells including B1 B cells, Langerhans cells, and microglia are not derived from adult HSCs, but arise during fetal hematopoiesis. Finally, they show that HSC contribution to multilineage hematopoiesis declines with aging. These results are consistent with the classical model of hematopoiesis in which HSCs continuously give rise to multi-lineage hematopoiesis through intermediate lineage-restricted progenitors and contrast with several recent reports challenging this model. The observation that the contribution to hematopoiesis declines with age is very interesting.

1) The authors highlight the recent publications on steady state mouse hematopoiesis by Sun et al. and Busch et al. which suggest that HSCs do not contribute significantly to multilineage blood production and that there can be long-term lineage restricted progenitors. The current results are in contrast to these other reports and the manuscript should highlight this distinction more clearly in the Abstract and Discussion.

2) It is a little surprising that the percentage of Tomato^+^ progenitors is not higher even after 28 weeks post-induction in MPPs as well as other committed progenitors. This phenomenon could be due to only a subset of HSCs differentiating at any given time, as has been suggested by others such as Camargo, but it also raises the possibility that the Tomato protein somehow alters HPSC function. Indeed, there is such a suggestion of this in Figure 1F in the long-term reconstitution experiments since grafts from Tomato positive and negative HSCs do not appear the same with respect to% reconstitution (or possibly even lineage composition). In order to show that Tomato does not cause problems with differentiation, the authors should show that the lineage potential and total cell output does not differ between Tomato^+^ and negative HSCs and sorted progenitors. Such studies can be performed in differentiation/reconstitution assays in vitro or in vivo, and ideally this should be performed using HSCs from multiple independent donor mice. Related to this point, how many independent donors were used in Figure 1F? It seems that at least 3 independent donors (biological replicates) would be required to answer the question of whether or not there are differences in the reconstitution potential or lineage potential of Tomato^+^ and Tomato^-^ HSCs.

3) Although the delayed contribution of old HSCs to MPPs is not surprising given that they are known to cycle slower than young HSCs, these experiments are only reliable if the Fgd5 reporter system works the same way in old HSCs as young; it does not appear the authors have confirmed this. In order to demonstrate this, the authors should show that Fgd5 expression is limited to HSCs in old mice and evaluate the distribution of Tomato labelling in the various types of HSCs (immunophenotypically defined My-bi, Balanced, Ly-bi) in old compared to young mice.

4) The authors conclude their studies indicate that adult HSCs do not contribute significantly to several lineages (e.g. microglial cells, Langerhans cells), and they therefore conclude that such cells arise from fetal HSCs. It is not clear from the Materials and methods exactly when the HSCs were induced to express Tomato, but it is likely in mice <8-10 weeks. Thus, while this speculation certainly is not unreasonable, at the same time these experiments were not designed in such a way to completely exclude this possibility. The authors are encouraged to be more circumspect with respect to the timing of the induction when making statements about whether or not post-natal HSCs may give rise to these cell populations. The Results and Discussion sections should be adjusted accordingly to mention this limitation of the studies.

---

## [Author Response]

[…] 1) The authors highlight the recent publications on steady state mouse hematopoiesis by Sun et al. and Busch et al. which suggest that HSCs do not contribute significantly to multilineage blood production and that there can be long-term lineage restricted progenitors. The current results are in contrast to these other reports and the manuscript should highlight this distinction more clearly in the Abstract and Discussion.

We have changed the wording in both the Abstract and Discussion to more directly emphasize our quite different conclusions compared to those of Sun et al. and Busch et al.

2) It is a little surprising that the percentage of Tomato^+^ progenitors is not higher even after 28 weeks post-induction in MPPs as well as other committed progenitors. This phenomenon could be due to only a subset of HSCs differentiating at any given time, as has been suggested by others such as Camargo, but it also raises the possibility that the Tomato protein somehow alters HPSC function. Indeed, there is such a suggestion of this in Figure 1F in the long-term reconstitution experiments since grafts from Tomato positive and negative HSCs do not appear the same with respect to% reconstitution (or possibly even lineage composition).In order to show that Tomato does not cause problems with differentiation, the authors should show that the lineage potential and total cell output does not differ between Tomato^+^ and negative HSCs and sorted progenitors. Such studies can be performed in differentiation/reconstitution assays in vitro or in vivo, and ideally this should be performed using HSCs from multiple independent donor mice. Related to this point, how many independent donors were used in Figure 1F? It seems that at least 3 independent donors (biological replicates) would be required to answer the question of whether or not there are differences in the reconstitution potential or lineage potential of Tomato^+^ and Tomato^-^ HSCs.

This is an insightful comment and we agree that the kinetic of replenishment of some multipotent progenitor subsets (especially GMLPs) is surprisingly slow. We interpret/discuss that this should reflect that some progenitors are imbued with a rather extensive self-renewal capability. A slow input to progenitor compartments from HSCs is also consistent with the preferential Tomato labeling of the least active HSCs by one Tamoxifen injection (Figure 1E).

While we don’t believe that the Tomato protein per se alters HSPC function (see Author response image 1), we however acknowledge that *Fgd5^CreERT2/+^* preferentially labels primitive/dormant HSCs after administration of lower doses of Tamoxifen. This is likely a consequence of higher Fgd5 expression in the most primitive HSCs (Säwén et al., 2016). We illustrate this in Figure 1E in our manuscript, which demonstrates preferential Tomato labeling by *Fgd5^CreERT2/+^* of the most H2B-label retaining HSCs. We consider dormancy/quiescence as a hallmark property of HSCs with the highest reconstitution potential, a view supported by a large number of previous studies. In light of this, we do not think it is surprising that Tomato^+^ HSCs could have somewhat different reconstitution patterns than Tomato negative HSCs (Figure 1F).

In an attempt to make a more fair comparison, we have compared the behavior of Tomato^+^ HSCs (in Figure 1F) to LSK-SLAM HSCs with a more restricted proliferation history (Säwén et al., 2016) following transplantation. This revealed a highly similar lineage distribution from Tomato^+^ HSCs and Tomato^-^ HSCs (see Author response image 1). One potential concern of this comparison is however the limited number of cells transplanted (5 cells and 10 cells, respectively, with rather limited numbers of mice/group). We therefore also compared the behavior of Tomato^+^ donor cells following reconstitution with 200 *Fgd5^CreERT2^* x Tomato HSCs (from Figure 4C in our manuscript) to a previous experiment in which we evaluated 100 HSCs isolated solely based on phenotype from WT mice (Supplementary information in Ugale et al., 2014). Also in this comparison, we observed no significant differences (2-tailed Student’s t-test) in lineage reconstitution between Tomato^+^ and Tomato^-^ HSCs. Collectively, we have therefore little reason to suspect altered cellular function caused by Tomato protein expression per se.

**Author response image 1. respfig1:** The Tomato fluorescent protein does not influence on the in vivo differentiation potential of HSCs.

3) Although the delayed contribution of old HSCs to MPPs is not surprising given that they are known to cycle slower than young HSCs, these experiments are only reliable if the Fgd5 reporter system works the same way in old HSCs as young; it does not appear the authors have confirmed this. In order to demonstrate this, the authors should show that Fgd5 expression is limited to HSCs in old mice and evaluate the distribution of Tomato labelling in the various types of HSCs (immunophenotypically defined My-bi, Balanced, Ly-bi) in old compared to young mice.

This is of course a highly relevant comment as a differential expression of Fgd5 in old animals could underlie our findings. We note that we in our original manuscript provided limited information on this. We began to address the HSCs specificity of the *Fgd5^CreERT2/+^* system in aged animals in Figure 4A, where we investigated the Tomato labeling of HSPCs after 5 daily tamoxifen injections. This showed that among LSK cells, virtually all Tomato^+^ cells are HSCs (or perhaps more accurately phrased, have an HSC phenotype) in old animals.

In addition to CreERT2, the mouse strain we work with also express a ZsGreen fluorescent reporter protein from the Fgd5-locus (Figure 2B). We have now taken advantage of this and compared Fgd5-ZsGreen expression in young and old mice (see Author response image 2). This revealed the highest Fgd5 expression in the CD150 highest HSCs (previously suggested to be myeloid-biased) both in young (8 weeks old) and aged (96 weeks old) mice, but also substantial expression of Fgd5 in CD150 intermediate HSCs (“Balanced”; in both young and old mice). A much more limited expression of Fgd5 is observed in CD150 negative candidate HSCs (Ly-bi). Compared to our previous work in which we defined HSC subtypes based on levels of CD150 (Beerman et al., 2010), we could here not include selection based on CD34-negativity (both because of technical incompatibilities but above all the well-established fact that levels of CD34 on HSCs fluctuate with activation). Not selecting for CD34-negativity likely underlies the differences in frequencies of these subsets in young/old mice (especially the larger fraction of CD150 negative cells) compared to our previous work (Beerman et al., 2010).

**Author response image 2. respfig2:** The expression of Fgd5 in HSC subsets of young and old mice.

4) The authors conclude their studies indicate that adult HSCs do not contribute significantly to several lineages (e.g. microglial cells, Langerhans cells), and they therefore conclude that such cells arise from fetal HSCs. It is not clear from the Materials and methods exactly when the HSCs were induced to express Tomato, but it is likely in mice <8-10 weeks. Thus, while this speculation certainly is not unreasonable, at the same time these experiments were not designed in such a way to completely exclude this possibility. The authors are encouraged to be more circumspect with respect to the timing of the induction when making statements about whether or not post-natal HSCs may give rise to these cell populations. The Results and Discussion sections should be adjusted accordingly to mention this limitation of the studies.

We thank the reviewers for this comment and completely agree that our original wording was inappropriate. Accordingly, we have made changes to the Results and Discussion:

Subsection title changed to “Fgd5-mediated HSC lineage tracing corroborates the fetal/juvenile origin of Langerhans cells, B1a B cells and brain microglial cells”.

Subsection “Fgd5-mediated HSC lineage tracing corroborates the fetal/juvenile origin of Langerhans cells, B1a B cells and brain microglial cells”: “This is in line with a more strict fetal/postnatal origin of B1a B cells.”

Discussion section: “Our work revealed robust HSC contribution to adult multilineage hematopoiesis and, while not focused at studying fetal hematopoiesis, support the previously reported fetal/juvenile origins of several other specific hematopoietic subsets.”

References:

Beerman I., Battacharya D., Zandi S., Sigvardsson M., Weissman I.L., Bryder D., Rossi D.J. Functionally distinct hematopoietic stem cells modulate hematopoietic lineage potential during aging by a mechanism of clonal expansion. *PNAS* March 23, 2010 107 (12) 5465-5470; https://doi.org/10.1073/pnas.1000834107